# Antioxidant Effects of Anthocyanin-Rich Riceberry™ Rice Flour Prepared Using Dielectric Barrier Discharge Plasma Technology on Iron-Induced Oxidative Stress in Mice

**DOI:** 10.3390/molecules26164978

**Published:** 2021-08-17

**Authors:** Natwalinkhol Settapramote, Niramon Utama-ang, Touchwin Petiwathayakorn, Kornvipa Settakorn, Saovaros Svasti, Somdet Srichairatanakool, Pimpisid Koonyosying

**Affiliations:** 1Division of Product Development Technology, Faculty of Agro-Industry, Chiang Mai University, Chiang Mai 50200, Thailand; natwalinkhol.se@rmuti.ac.th (N.S.); niramon.u@cmu.ac.th (N.U.-a.); 2Cluster of High Value Product from Thai Rice for Health, Chiang Mai University, Chiang Mai 50200, Thailand; 3Department of Agro-Industry, Faculty of Agriculture and Technology, Surin Campus, Rajamangala University of Technology Isan, Surin 32000, Thailand; 4Department of Biochemistry, Faculty of Medicine, Chiang Mai University, Chiang Mai 50200, Thailand; touchwinchalae@gmail.com (T.P.); stkornvipa@gmail.com (K.S.); somdet.s@cmu.ac.th (S.S.); 5Thalassemia Research Center, Institute of Molecular Biosciences, Mahidol University, Nakorn Pathom 71300, Thailand; stssv@yahoo.com

**Keywords:** anthocyanins, antioxidant, iron overload, plasma technology, reactive species, Riceberry™ rice

## Abstract

Redox-active iron generates reactive oxygen species that can cause oxidative organ dysfunction. Thus, the anti-oxidative systems in the body and certain dietary antioxidants, such as anthocyanins, are needed to control oxidative stress. We aimed to investigate the effects of dielectric barrier discharge (DBD) plasma technology in the preparation of Riceberry™ rice flour (PRBF) on iron-induced oxidative stress in mice. PRBF using plasma technology was rich in anthocyanins, mainly cyanidine-3-glucoside and peonidine-3-glucoside. PRBF (5 mg AE/mg) lowered WBC numbers in iron dextran (FeDex)-loaded mice and served as evidence of the reversal of erythrocyte superoxide dismutase activity, plasma total antioxidant capacity, and plasma and liver thiobarbituric acid-reactive substances in the loading mice. Consequently, the PRBF treatment was observed to be more effective than NAC treatment. PRBF would be a powerful supplementary and therapeutic antioxidant product that is understood to be more potent than NAC in ameliorating the effects of iron-induced oxidative stress.

## 1. Introduction

In mammalian cells, mitochondrial oxidative phosphorylation produces adenosine triphosphate as a high-energy compound [1], along with the superoxide radical (O_2_^−•^) as a by-product. Consequently, reactive oxygen species (ROS) are capable of destroying biomolecules leading to oxidative cell damage [2,3]. ROS and other reactive species are fundamental for living systems, without them organisms would perish. On the other hand, an excess of antioxidants may be detrimental and lead to a condition known as prooxidant. In this case, the body produces certain anti-oxidative enzymes, such as superoxide dismutase (SOD), catalase (CAT) and glutathione peroxidase (GPx), along with nucleophilic substrates such as glutathione (GSH), thioredoxin and a reduced form of nicotinamide adenine dinucleotide phosphate (NADPH), all of which are needed to balance the levels of harmful free radicals [3,4]. In addition, external factors including high caloric diets (such as those involving high sugar, fat and alcohol intake), as well as exposure to air pollution and radiation, can generate ROS that can overwhelm antioxidant defenses leading to oxidative stress [4,5,6]. The long-term contribution of oxidative stress can cause certain chronic and degenerative diseases including cancer, diabetes, cardiovascular diseases and Alzheimer’s disease [1,5,6,7,8,9,10]. Accordingly, the elevation of antioxidant levels and the elimination of ROS generation are suggested to minimize oxidative stress [11]. Under iron-overloaded conditions, redox-active forms of iron can non-enzymatically catalyze ROS generation, which can then lead to organ dysfunction and eventually death. Moreover, certain other natural products, in conjunction with iron chelators, can relieve oxidative organ damage [12,13,14,15,16,17,18,19,20,21]. Furthermore, dietary antioxidants present in fruits and vegetables play important roles in disease prevention, radical scavenging and oxidative activation of the nuclear factor erythroid 2–related factor 2 in the signaling pathway [22].

Up to now, more than 500 anthocyanins (ACNs) have been distributed and identified in plants (80% in the leaves, 69% in the fruits and 50% in the colored flowers). These were originated by cyanidin, delphidin and pelargonidin, and their light absorption was recorded at about 500 nm [23]. Impotantly, the roles of ACNs in plants were initiated in response to biotic and abiotic stress, for which *O. sativa* synthesizes ACNs in the response to water, salt and fungi stresses [23]. Nowadays, many ACNs-rich supplementary products formulated from edible fruits and vegetables exert preventive and therapeutic effects on various human diseases [24,25,26,27,28,29]. Riceberry™ rice (RB, *Oryza sativa*) was developed in 2002 from the cross-breeding of Thai Jao Hom Nin purple rice and Thai Khoa Dawk Mali 105 rice by Professor Dr. Apichart Vanavichit’s research team at the Rice Science Center, Kasetsart University, Thailand. The trademark for this type of rice was granted in 2012. Importantly, the purple-black colored Riceberry™ grain is abundant in antioxidants, predominantly ACNs and anti-inflammatory agents that can be used to treat a number of diseases [30]. Considerably, RB flour is rich in phenolic compounds, particularly ACNs (188 ± 22 mg/g), of which cyanidin-3-glycosides (C3G) are the major compounds (50.32 ± 2.01 mg C3G/g) when compared with rice flour (0.10 ± 0.05 mg C3G/g) [31,32]. In addition, ACNs concentrations present in brewed RB vinegar, fermented RB vinegar and wine were reported to be 1.62, 10.92 and 18.73 mg/mL, respectively [33,34]. From a chromatographic analysis, RB extract contained four phytosterols (e.g., 24-methyleneergosta-5-en-3β-ol, 24-methyleneergosta-7-en-3β-ol, fucosterol and gramisterol), three triterpenoids (e.g., cycloeucalenol, lupenone and lupeol) and three main phenolic compounds (e.g., protocatechuic acid, vanillic acid and anthocyanin) [32,35,36]. Importantly, total phenolic and total anthocyanin contents were found to be approximately 500 times higher in RB flour than in rice flour [32].

Likewise, the anti-oxidative properties of RB can be increased using the dielectric barrier discharge (DBD) plasma technology, which is a non-thermal process that involves the application of a high-frequency electric field in the gaseous state [37]. By way of its function, DBD plasma technology results in an effective distribution of a given substance on the surface area of products [38,39]. In the food production industry, plasma technology has been utilized in the preparation of wheat flour, rice flour, corn flour and tapioca flour [40,41,42,43,44,45]. It has been employed to increase bioactive phytochemical yields, as well as antioxidant activity, and to inhibit the growth of certain microorganisms [46,47,48,49,50]. In terms of its potential health benefits, pureed RB pudding subjected to plasma technology resulted in lower glycemic index values than the liquidized original products, while the bioavailability values recorded in healthy volunteers were not found to be different [51]. This study aimed to use DBD plasma technology in the preparation of Riceberry™ rice flour (PRBF) and to investigate the effects on iron-induced oxidative stress in mice.

## 2. Results

### 2.1. Anthocyanin Content of Riceberry™ Rice Flour (PRBF) Using Plasma Technology

Anthocyanin content in PRBF was determined using the reverse-phase HPLC method. ACNs content in PRBF was found to be 1.94 mg/g, while the main ACNs were C3G and P3G (Figure 1). The anthocyanin content of PRBF using DBD plasma technology related to previous study [3]. Two major compounds, cyanidin-3-glucoside (C3G) (peak a) and peonidin-3-glucoside (P3G) (peak b), were identified by comparison with the retention times of the two standards and the concentrations in terms of mg anthocyanin equivalent (AE) were calculated.

### 2.2. Effect of PRBF Treatment on BW and OWI

In both preventive and therapeutic studies, BW values in all groups of mice with and without and FeDex loading for 14 d were not found to have significantly changed, while NAC (100 mg/kg) and PRBF (5 mg AE/kg) treatments over 14 d did not alter the BW values in the mice of all study groups (Appendix A). In the preventive study, weight indices of the liver, heart, pancreas and spleen (OWI) taken from mice with or without FeDex loading, as well as from the FeDex-loaded mice treated with NAC and PRBF, revealed no significant differences (Appendix A). As was found in the therapeutic study, weight indices of the heart, pancreas and spleen were not different in subjects of all study groups; nonetheless, the liver weight index of the FeDex-loaded mice was found to be less than that of the mice without FeDex loading. Accordingly, this value was found to be lower following the NAC and PRBF treatments (Appendix A).

### 2.3. Effect of PRBF Treatment on Hematopoiesis

WBC numbers were found to have increased significantly in mice with FeDex loading in both preventive and therapeutic studies when compared to mice without FeDex loading, while the increase in WBC was reversed by the NAC and PRBF treatments in the preventive study and not in the therapeutic study (Figure 2A). However, the WBC numbers in all the groups remained within a normal range. Furthermore, levels of RBC numbers, Hb concentrations and PLT numbers were not found to be significantly different among all groups in the two study models (Figure 2B–D). This was true except for the PLT numbers in iron-loaded mice that had been treated with NAC in the therapeutic study. The PLT numbers in mice in this group were higher than in the other groups.

### 2.4. Effect of PRBF Treatment on Erythrocyte SOD Activity

According to our findings, Fe-Dex loading significantly decreased erythrocyte SOD activity in mice in both the preventive study and therapeutic study (Figure 3). Treatments with PRBF (5 mg AE/g) and NAC (100 mg/kg) equally restored SOD activity in FeDex-loaded mice in the preventive study. In comparison, only the NAC treatment also restored SOD activity, whereas PRBE did not produce the same results in the therapeutic study.

### 2.5. Effect of PRBF on Plasma TAC

When excessive iron generates high amounts of ROS leading to oxidative stress in the body, anti-oxidative compounds and enzymes can efficiently destroy the persisting ROS. All treatments in the preventive study did not influence TAC levels in plasma collected from FeDex-loaded mice (Figure 4). In terms of the therapeutic study, FeDex-loaded mice revealed lower plasma TAC levels than mice without FeDex loading (*p* < 0.05); however, both PRBF and NAC treatments were able to restore a decrease in the plasma TAC levels but not to a significant degree (Figure 4). Thus, antioxidant PRBF may be rapidly cleared from the plasma or metabolized in the liver, while the compound should be supplemented continuously to sustain effective concentrations in the body.

### 2.6. Effect of PRBF on Plasma and Liver TBARS

In this study, TBARS concentrations representing lipid-peroxidation products were measured in the plasma and liver of mice in both the preventive and therapeutic studies, for which the results are shown in Figure 5 and Figure 6, respectively. Results from the preventive study (Figure 5) have revealed that FeDex loading was found to have increased plasma TBARS levels significantly when compared to the treatments of mice without FeDex loading, while only PRBF (5 mg AE/kg) treatment lowered the increase in plasma TBARS levels. Similarly, in the therapeutic study, FeDex loading resulted in a significant increase in plasma TBARS levels, whereas PRBF and NAC (100 mg/kg) were found to have reduced the increased TBARS levels in the plasma (Figure 5). In addition, the plasma TBARS levels observed in the preventive study were reported to be even higher than those in the therapeutic study.

Consistently, TBARS concentrations in the livers of FeDex-loaded mice in the preventive study were found to be significantly increased when compared to those concentrations in mice without FeDex loading, while the concentrations were decreased in treatments with PRBF (5 mg AE/kg) (*p* < 0.05) and NAC (100 mg/kg) (Figure 6). Likewise, in the therapeutic study, TBARS concentrations in the livers of FeDex-loaded mice were significantly increased when compared to mice in treatments without FeDex loading; however, the PRBF and NAC treatments were able to significantly lower the increase in hepatic TBARS concentrations (Figure 6). Taken together, PRBF can effectively inhibit ROS-induced lipid-peroxidation reactions in iron-loaded plasma and livers, for which the activity would be more effective than for the reference antioxidant NAC.

## 3. Discussion

Pigmented rice (e.g., black, red, brown, dark purple and Riceberry™ rice) contain flavones, ACNs, proanthocyanidins, tannins, phenolics, γ-oryzanols, tocopherols, phytosterols and essential oils that are known to be beneficial to human health [52,53]. According to the chromatographic analysis, proanthocyanidins were only detected in red rice while ACNs were detected in black and purple rice, wherein they contributed to antioxidant activity [53]. Anthocyanins, a group of hydrophilic flavonoids, are natural colorants responsible for the red and purple color of pigmented rice, other cereal grains, and fruits and vegetables. They are also involved in certain other important biological activities [54]. Our present study has revealed that purple-black colored Riceberry™ rice flour comprises two major components of ACNs: C3G and P3G. Consistently, high-performance-liquid chromatographic/mass spectrometric and electron paramagnetic resonance imaging analyses have been used to identify major functional ACNs including C3G, P3G, cyanidin 3-galactoside, cyanidin 3-rutinoside, cyanidin-3,5-diglucoside, malvidin 3-galactoside and pelargonidin-3,5-diglucoside in pigmented rice, in which C3G and P3G are known to be the predominant components of the ACNs [52,55,56]. Notably, ACNs that are persistent in certain plant foods, such as pigmented rice, are unstable in the small intestines and can only be absorbed with low bioavailability. Additionally, they may contribute to a range of poor bioactivities; therefore, suitable rice processing is required. The processing methods include solvent extraction, acidification, irradiation, fermentation, germination, soaking, boiling, baking, extrusion and plasma technology; nonetheless, they may actually change the profile and yields of ACNs and other phenolic compounds, or even influence certain other biological activities [57]. For instance, thermal cooking was found to decrease total ACNs and C3G contents and antioxidant activity but did not influence anti-inflammatory activity in black rice [58]. Treatment of Khao Dawk Mali 105, Hom Nil, KiawNgu and LeumPua rice extracts at 100 °C for 15 min degraded phenolic components and total phenolic content (TPC) but did not change their antioxidant activities. However, acidification of the extracts did increase TPC and antioxidant activity values [59]. Likewise, a microwave cooking method could markedly decrease phenolics, ACNs contents and antioxidant activities in purple rice when compared to uncooked rice, while an electric rice cooker, an autoclave and a microwave oven method all resulted in a decrease in C3G content and an increase in protocatchuic acid, along with an anti-proliferation effect on Caco-2 cells [60].

Nowadays, consumers demand safe, high-quality, nutritious ACNs-rich pigmented foods (e.g., red clover, wheat and Riceberry™ rice) owing to their health benefits. These foods can also be utilized as pharmaceutical ingredients. Importantly, they serve as free-radical scavengers and possess antioxidant, anti-inflammatory and anti-microbial properties [24,25,26,27,61]. Cold plasma or dielectric barrier discharge (DBD) plasma technology is a non-thermal technique involving atmospheric pressure that has been developed to decontaminate the surfaces of medical instruments, living tissues and food products to verify that they are free from water, chemicals and microorganisms. By using high-voltage electrical discharges, microwaves and gas ionization processes, as well as plasma technologies, can efficiently kill or inactivate bacteria, yeasts, fungi, pathogens, spores and biofilms. Unfortunately, a 23% loss of ACNs was found in plasma-treated chokeberries, while a 9% increase in ACNS was found in pasteurized (80 °C, 1 min) chokeberries [62]. In addition, the total aerobic plate count in blueberries with plasma technology was significantly lowered when compared to blueberries without the treatment, whereas total ACNs content (mainly C3G) was significantly decreased after 90 s of the plasma treatment. This outcome was probably due to a degradation of ACNs at temperatures >38 °C [63]. On the contrary, blueberry juice with plasma technology resulted in increases in phenolics and ACNs contents, as well as antioxidant activity [64]. Likewise, Thai geminated brown rice cultivars exposed to the plasma technology produced higher contents of ACNs, total phenolic compounds, certain simple phenolics, γ-oryzanols, tocopherols, phytosterols and triterpenoids, but did not change the antioxidant activity when compared to untreated rice [48]. Moreover, pomegranate juice exposed to plasma treatment technology (3 min) yielded a 14% increase of ACNs content when compared to untreated juice [65]. Therefore, we utilized cold plasma for the treatment of Riceberry™ rice flour to achieve high ACNs yields as well as antioxidant capacity. Our previous study proved that dielectric barrier discharge (DBD) plasma technology can improve antioxidant activity and anthocyanin content of Riceberry rice flour [32]. This was also done to sterilize the Riceberry™ rice product for further studies involving animals.

Hematological parameters, like WBC numbers, can be used to indicate FeDex-induced oxidative stress since they are significantly increased when compared to subjects without iron loading [66]. Under blood oxidative stress, increased WBC and PLT numbers generated high plasma levels of ROS, which then provided additional protection against infection and pathogen migration [67]. In addition, acute ethanol stress significantly increased total WBC numbers (particularly neutrophils), along with ROS and malondialdehyde productions in mice. This outcome was possibly due to an increase in liver microsomal cytochrome P450 detoxification, while melatonin restored an appropriate increase in these oxidative stress biomarkers [68]. In controversy, Schumann and colleagues have previously demonstrated that WBC, erythrocyte SOD and CAT values, along with serum C-reactive proteins, total oxidative capacities and TBARS values, did not respond to the oral dosing of 120 mg ferrous sulfate/day over 7 days [69]. In our study, we found that FeDex loading (10 mg each) significantly increased WBC numbers in C57BL/6 mice, indicating an iron-induced inflammatory condition; however, ACNs-rich PRBF treatment lowered the increase in WBC numbers but did not influence other blood cell populations.

With regard to the antioxidant defense in the body, SOD, TAC and TBARS were used as key biomarkers. Serum and cellular SOD activities can be decreased by an overrun in ROS in oxidative stress in conjunction with an increase in antioxidant supplementation to couple O_2_^−•^ and 2H^+^. This was done to form H_2_O_2_ before a conversion to nontoxic water [70]. TAC also indicates the antioxidant status possessed by certain antioxidant compounds (e.g., GSH, NADPH, phenolic compounds and NAC) and enzymes (CAT, GPx and SOD) in cells and plasma compartments [71,72]. In addition, TBARS, such as MDA, are measurable final products of lipid-peroxidation reactions and are commonly used as oxidative stress biomarkers. Herein, PRBF treatment was found to improve erythrocyte SOD activity and efficiently restore plasma TAC of FeDex-loaded mice. FeDex loading enhanced the production of ROS that would subsequently initiate chain-reaction peroxidations of plasma and cellular polyunsaturated fatty acids resulting in an increase in TBARS concentrations. In the therapeutic study, plasma TBARS concentrations in FeDex-loaded mice were found to be lower than those obtained from the preventive study, while the liver TBARS concentrations were found to be inversed and vice versa. Possibly, mice used in the preventive study were exposed to FeDex for only 24 h before being euthanized, for which most of the iron remained in the plasma and was accessed to catalyze the peroxidation of plasma lipoproteins rapidly. This ultimately resulted in higher plasma TBARS concentrations. Meanwhile, mice used in the therapeutic study were exposed to FeDex continuously for 14 d, for which most of the iron was cleared from the plasma and taken up into liver cells. Eventually, the iron was intently presented in the liver rather than in the plasma and utilized for functional molecules or deposited in ferritin molecules [73,74]. Consistent with our study, consumption of yogurt supplemented with ACNs-rich Riceberry™ rice rapidly increased levels of TAC and MDA in the plasma of healthy subjects when compared with the control yogurt [75]. In another study, consumption of ACNs-rich Riceberry™ rice bread significantly increased TAC levels in the plasma collected from heathy subjects, but did not alter their MDA levels [76]. Hou and colleagues have reported a preventive effect of ACNs-rich black rice extract (500 mg/kg) on ethanol-induced liver inflammation in male Wistar rats by significantly lowering the increases of liver enzyme activities in the serum, as well as MDA concentrations in the serum and liver [77]. Accordingly, the consumption of black-rice ACNs (200 mg/kg) decreased the body weight of C57BL/6 mice fed with a high-fat diet by 9.6%, while also decreasing hepatic lipid-peroxidation products, increasing hepatic SOD and GPx activities, and resulting in a downregulation in the expression of tumor necrotic factor-alpha, interleukin-6, inducible nitric oxide synthase and nuclear factor-kappa B genes. All of which were indicative of an alleviation of oxidative stress and inflammation [78]. Importantly, consumption of ACNs-rich black rice extract could lower plasma TBARS and blood oxidized glutathione concentrations in high fructose diet-fed rats [79]. Furthermore, C3G and P3G, as the two main components of ACNs in black rice extract, exerted antioxidant and anti-inflammatory properties in vitro and murine macrophage RAW264.7 cells [56]. Similarly, ACNs-rich Riceberry™ bran extract was found to reduce the increase of hepatic MDA concentrations and up-regulate the expression of SOD genes in gentamycin-induced hepatotoxicity in rats. This outcome is supportive of the free-radical scavenging antioxidant, iron-chelating, anti-inflammatory and anti-apoptotic properties of the Riceberry™ extract [80]. Importantly, the phytochemicals present in vegetables and fruits function as free radical scavenger antioxidants and can also induce microsomal phase II enzymes. This is known to lead to the induction of the nuclear factor erythroid 2-related factor 2/electrophilic responsive element (Nrf2/EpRE) signaling system [22]. Notably, a black rice diet that is rich in ACNs, in particular wiith regard to C3G- and protocatchuic acid-rich alleviated ferric-nitrilotriacetate-induced rats, indicated that renal GPx activity was effectively decreased, whereas SOD, glutathione S-transferase and NAD(P)H quinone reductase were not associated with lipid peroxidation levels [81]. Likewise, blueberry and cranberry juices, as well as cyanidin, decreased ROS production and lipid peroxidation in neuroblastoma SH-SY5Y cells induced by hydrogen peroxide by upregulating SOD and CAT activities [82]. More importantly, ACNs, including cyanidin, delphinidin, delphinidin-3-glucoside, malvidin, petunidin and petunidin-3-glucoside, were found to release iron (Fe^2+^) from soybean seed ferritin, which strongly suggests their iron-chelating activity [83].

## 4. Materials and Methods

### 4.1. Chemicals and Reagents

All organic solvents used in this study were of the highest pure grade or HPLC grade. 2,2′-Azino-bis(3-ethylbenzothiazoline-6-sulfonic acid) diammonium salt (ABTS), butyrated hydroxyl toluene (BHT), cyanidine-3-glucoside (C3G), dimethyl sulfoxide (DMSO), iron dextran (FeDx), N-acetylcystein (NAC), peonidin-3-glucoside (P3G), sodium dodecyl sulphate (SDS), 1,1,3,3-tetramethoxypropane (TMP), thiobarbituric acid (TBA), 6-hydroxy-2,5,7,8-tetramethylchroman-2-carboxylic acid (Trolox) and 2-(4-iodophenyl)-3-(4-nitrophenyl)-5-(2,4-disulfophenyl)-2*H*-tetrazoliumand monosodium salt (WST-1) were all obtained from Sigma-Aldrich Chemicals (St. Louis, MO, USA). Normal-formula food (C.P. MICE FEED, Serial Number 082) was purchased from Perfect Companion Group Company Limited (Bangsaothong District, Samutprakarn Province, Thailand) and was nutritionally composed of metabolic energy (3040 kcal/kg), crude protein (24%), fat (4.5%), fiber (5%), calcium (1.0%), phosphorus (0.9%), sodium (0.20%), potassium (1.17%), magnesium (0.23%%), manganese (171 parts per million or ppm), copper (22 ppm), zinc (100 ppm), iron (180 ppm), cobalt (1.82 ppm), potassium iodide (1 ppm), selenium (0.1 ppm), vitamin A (20,000 IU/kg), vitamin D (4000 IU/kg), vitamin E (100 mg/kg), vitamin K (5 mg/kg), vitamin B1 (20 mg/kg), vitamin B2 (20 mg/kg), vitamin B6 (20 mg/kg), vitamin B12 (0.0036 mg/kg), niacin (100 mg/kg), folic acid (6 mg/kg), biotin (0.4 mg/kg), pantothenic acid (60 mg/kg) and choline chloride (1500 mg/kg).

### 4.2. Methods

#### 4.2.1. Production of Riceberry Rice Flour Using Plasma Technology

Riceberry™ rice grains were purchased from a local grocery store located in Mueang Pan District (Lampang Province, Thailand). The rice grains were milled into flour and treated with a DBD plasma generator (Model: PLASMA TAC, ADTEC Plasma Technology Company Limited, Hounslow, UK) under the following conditions: an operating time of 7.87 min, power set at 166 watts, 0.64 L/min of oxygen gas, 16 L/min of argon gas and 10 L/min of nitrogen gas to achieve PRBF [32]. In the assay, PRBF was dissolved in deionized water (DI) (1:2, *w*/*w*) and a suspension of the starch was heated until it became completely gelatinized. Subsequently, α-amylase (11,000 U/mL) was then added and the mixture was continuously stirred for 30 min at 40 °C. The mixture was stirred one more time for 10 min at 50 °C to stop the enzyme activity. The slurry was then placed in a freeze-drying machine. Lastly, the dried flour was weighed and kept at 4 °C for further use.

#### 4.2.2. HPLC Analysis of Anthocyanin Content

Anthocyanin content in PRBF was determined using the reverse-phase HPLC method [84]. Anthocyanin solution (1 mg/mL) was prepared in 1% DMSO and the sample (10 µL) was analyzed using the HPLC system(Agilent Technologies, Santa Clara, CA, USA) employing a C18 type column (250 mm × 250 mm, 5 μm particle size, Agilent Technologies) a regulated temperature of 30 °C and a programmed mobile-phase involving water, methanol and formic acid (75:18:7, *v*/*v*/*v*) with isocratic elution at a flow rate of 0.5 mL/min that was utilized to detect ACNs at a wavelength of 510 nm [28]. Two selected ACNs, namely C3G and P3G, were compared with C3G and P3G standards and their concentrations were calculated using the standard curves.

#### 4.2.3. Animal Care

Male wild type mice (Strain C57BL/6, aged 2–3 m, body weight 20–25 g) were kindly provided by the Thalassemia Research Center, Institute of Molecular Biosciences, Mahidol University Salaya Campus (Nakorn Pathom, Thailand). Mice were housed in non-metallic cages in a clean room under standard conditions (22 ± 1 °C and 40–60% humidity, 12-h day/12-h night cycle) in the Animal House Unit of the Faculty of Medicine, Chiang Mai University. The Animal Study Protocol was approved of by the Animal Ethical Committee of the Medical Faculty, Chiang Mai University, Thailand (Protocol Number 26/2562). All animal experiments were carried out in accordance with the guidelines of the National Institutes of Health in terms of the care and use of laboratory animals (NIH Publications No. 8023, revised 1978).

#### 4.2.4. Iron Loading and PRBF Treatment in Mice

All mice were given free access to clean drinking water ad libitum and normal diets. The body weight (BW) of each mouse was recorded every four days. In the preventive study, mice were randomly divided into 4 groups (*n* = 6 each): group 1 received DI and groups 2–4 were fed DI, NAC (200 mg/kg) and PRBF (5 mg AE/kg), respectively using a gavage needle once a day for 14 d. Two hours after the last administration, normal saline solution (NSS) (100 µL) was intraperitoneally (ip) administrated to control group 1, while 10 mg/100 µL Iron dextran (FeDex) were ip administrated to experimental groups 2–4. The mice were euthanized (sacrificed) by cervical dislocation on day 15, when heart blood was collected for hematological and biochemical analyses. Internal organs, including the liver, heart, pancreas and spleen, were removed and kept frozen at −80 °C for biochemical analysis as has been described below.

In the therapeutic study, mice were randomly divided into 4 groups (*n* = 6 each): group 1 was daily given DI along with an ip injection of NSS for 14 d; meanwhile, groups 2, 3 and 4 were orally administrated DI, NAC (200 mg/kg), and PRBF (5 mg AE/kg) along with an ip administration of FeDex (10 mg) for 14 d. On day 15, the mice were euthanized by cervical dislocation and the heart blood was collected for hematological and biochemical analyses. Internal organs were excised, weighed and kept frozen at −80 °C for biochemical analysis as has been described below. In addition, the organ weight index (OWI) was determined as a ratio of organ weight to body weight (BW).

#### 4.2.5. Hematological Parameter Analysis

Complete blood count (CBC), including white blood cell (WBC) numbers, red blood cell (RBC) numbers, Hb concentration, Hct, mean corpuscular volume (MCV), mean corpuscular hemoglobin (MCH), mean corpuscular hemoglobin concentration (MCHC), platelet (PLT) numbers and reticulocyte numbers, were determined using an automatic cell counter at the Central Laboratory, Faculty of Veterinary Medicine, Chiang Mai University, Thailand according to the manufacturer’s instructions.

#### 4.2.6. SOD Activity Assay

Superoxide dismutase (SOD) is known to catalyze a dismutation of superoxide anion into hydrogen peroxide and molecular oxygen. SOD activity was determined using WST-1 tetrazolium salt as an electron acceptor that was used to reduce the solution to a water-soluble WST-1 formazan product giving the absorption spectrum at 440 nm [85]. Herein, SOD activity in RBC hemolysate was measured using a commercially available SOD determination kit (Catalogue number 19160, Sigma-Aldrich Chemie, Buchs, Switzerland) according to the manufacturer’s instructions. Briefly, hemolysate and DI (20 µL) were mixed with the WST working solution (200 µL), then incubated with a xanthine/xanthine oxidase working solution at 37 °C for 20 min and the absorbance (A) was measured at 440 nm using a 96-well microplate reader. Since the absorption of WST-1 formazan was directly proportional to the amount of superoxide anion, SOD activity (% inhibitory rate) was calculated using the following equation: % Inhibition of SOD activity = 100 × (A_hemolysate_ − A_DI_)/A_DI_

#### 4.2.7. Determination of Plasma Antioxidant Capacity

Antioxidant capacity (AC) was assayed based on decolorization of blue-colored ABTS^•+^ cationic radicals. Briefly, working ABTS^•+^ solution (1 mL) was incubated with plasma or standard Trolox solution (0.01 mL) for exactly 6 min and A values were immediately measured at 734 nm against the reagent blank. Plasma AC was calculated from the standard curve of Trolox and is expressed as Trolox-equivalent antioxidant capacity (TEAC).

#### 4.2.8. Quantification of Plasma and Liver TBARS

Concentrations of thiobarbituric acid-reactive substances (TBARS) were measured using the fluorometric method established by Masowicz and colleagues with slight modifications [14]. Briefly, plasma, tissue homogenate and standard TMP solution (80 µL) were mixed with 0.2% BHT (10 µL). Next, 0.44 M metaphosphoric acid (240 µL) and 0.6% (*w*/*v*) TBA (160 µL) were added to the mixture, which was then incubated at 90 °C for 30 min and cooled down in an ice bath at 4 °C for 10 min. After that, *n*-butanol (300 µL) was added to the subsequent mixture and it was centrifuged at 3000 rpms for 15 min. Finally, the A values of the supernatant were measured using a microplate reader at a wavelength of 540 nm.

#### 4.2.9. Statistical Analysis

Data were analyzed using Statistical Package for the Social Sciences (SPSS) version 17.0 (IBM Corporation, Armonk, NY, USA) and are shown as mean ± standard deviation (SD) values or mean ± standard error values of the mean (SEM). An independent Student’s-test was used to determine statistical significance, for which *p* < 0.05 was considered indicative of a significant difference.

## 5. Conclusions

Together with the plasma technology, cyanidin-3-glucoside and peonidine-3-glucoside are the two main components of anthocyanins that are abundant in PRBF and present potent antioxidant and anti-lipid peroxidationtoa better degree than N-acetylcystein. This can lead to an amelioration of oxidative stress in iron-induced mice. Moreover, insoluble bound fractions of other phenolics, such as p-coumaric acid, ferulic acid, isoferulic acid and vanillic acid that are present in Riceberry™ rice, may contribute to the alleviation of oxidative stress. In future work, Riceberry™ rice products with DBD plasma technology would be used in the formulation of nutraceuticals or functional foods. Furthermore, they would be employed in clinical investigations involving oxidative-stress patients.

## Figures and Tables

**Figure 1 molecules-26-04978-f001:**
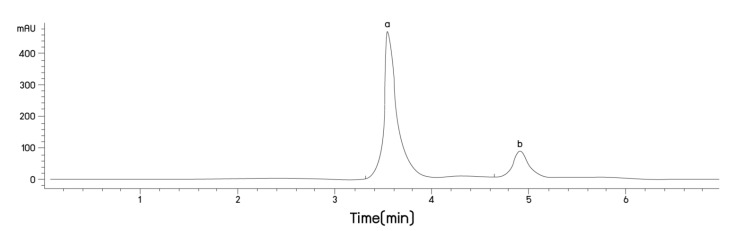
HPLC analysis of C3G and P3G in PRBF product. Plasma-treated Riceberry™ rice flour (PRBF) was analyzed using the HPLC conditions described in Section 2.3. Two major compounds, cyanidin-3-glucoside (C3G) (peak a) and peonidin-3-glucoside (P3G) (peak b), were detected at 510 nm and identified by comparison with the retention times of the two standards and the concentrations in terms of mg anthocyanin equivalent (AE) were calculated.

**Figure 2 molecules-26-04978-f002:**
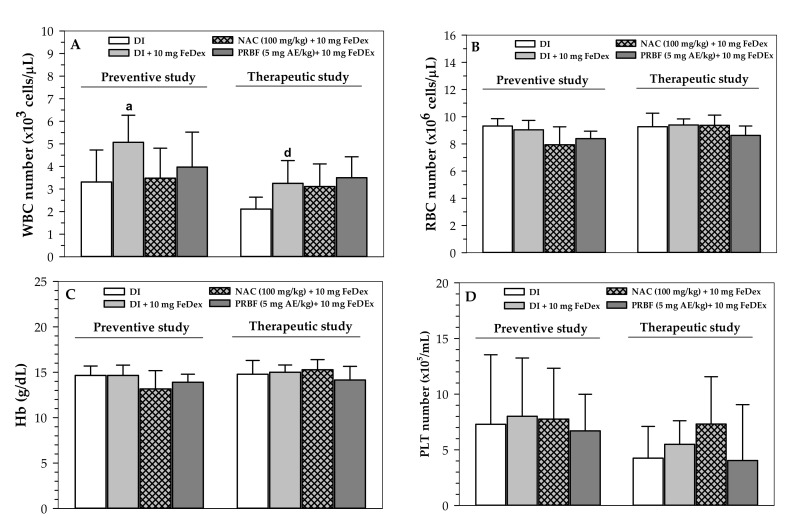
Hematological parameter levels in FeDex-loaded mice treated with DI, NAC and PRBF. In terms of a preventive effect, mice were treated with DI, NAC (100 mg/kg) and PRBF (5 mg AE/kg) for 14 d. This was followed by a single ip injection with 10 mg FeDex after the last oral dose was administered on day 14. In terms of a preventive effect, mice were then sacrificed; their heart blood was collected and their hematological parameter levels were determined. In term of a therapeutic effect, mice were ip injected with 10 mg FeDex together with DI, NAC (100 mg/kg) and PRBF (5 mg AE/kg) for 14 d. Mice were then sacrificed, their heart blood was collected and their hematological parameter levels were determined. Data are expressed as mean ± SD values (*n* = 6). ^a^
*p* < 0.05 when compared with DI alone in the preventive study; ^d^
*p* < 0.05 when compared with DI alone in the therapeutic study. Abbreviations: AE = anthocyanin equivalent, PRBF = plasma-treated Riceberry™ rice flour, DI = deionized water, FeDex = iron dextran, Hb = hemoglobin (**C**), NAC = *N*-acetylcystein, PLT = platelets (**D**), RBC = red blood cells (**B**), WBC = white blood cells (**A**), SD = standard deviation.

**Figure 3 molecules-26-04978-f003:**
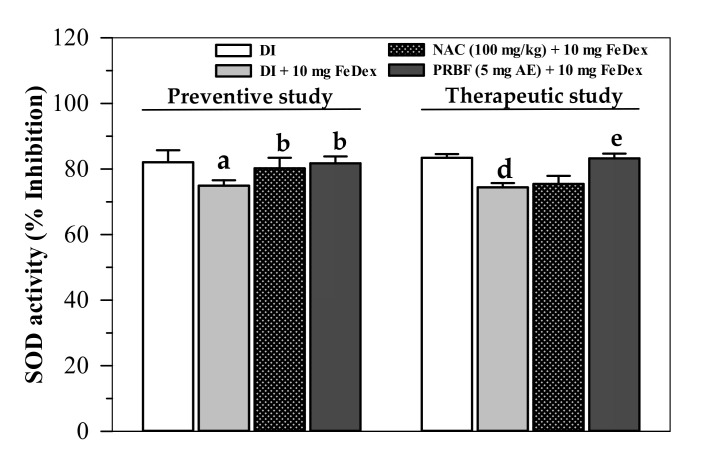
Inhibition of erythrocyte SOD activity in FeDex-loaded mice that had been treated with DI, NAC and PRBF. In the preventive study, mice were treated with DI, NAC (100 mg/kg) and PRBF (5 mg AE/kg) for 14 d. This was followed by a single ip injection with 10 mg FeDex after the last oral dose was administered on day 14. Mice were then sacrificed; their heart blood was collected and their erythrocyte SOD activity was determined. Results are presented in an inhibition graph. In the therapeutic study, mice were ip injected with 10 mg of FeDex together with DI, NAC (100 mg/kg) and PRBF (5 mg AE/kg) for 14 d. Mice were then sacrificed, their heart blood was collected and erythrocyte SOD activity was determined. Data are expressed as mean ± SD values (*n* = 6). ^a^
*p* < 0.05 when compared with DI and ^b^
*p* < 0.05 when compared with DI in FeDex loaded mice in the preventive study; ^d^
*p* < 0.05 when compared with DI and ^e^
*p* < 0.05 when compared with DI in Fe-Dex loaded mice in the therapeutic study. Abbreviations: AE = anthocyanin equivalent, PRBF = plasma-treated Riceberry™ rice flour, DI = deionized water, FeDex = iron dextran, NAC = *N*-acetylcystein, SD = standard deviation, SOD = superoxide dismutase.

**Figure 4 molecules-26-04978-f004:**
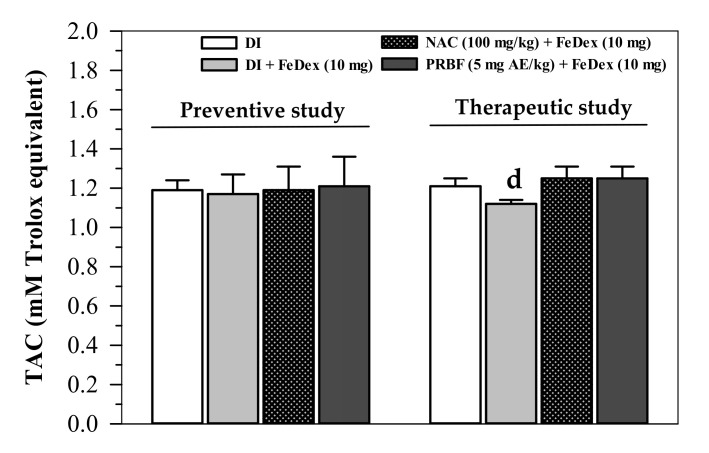
Levels of TAC in the plasma collected from FeDex-loaded mice that had been treated with DI, NAC and PRBF. In the preventive study, mice were treated with DI, NAC (100 mg/kg) and PRBF (5 mg AE/kg) for 14 d. This was followed by a single ip injection of 10 mg FeDex after the last oral dose was administered on day 14. Mice were then sacrificed; their heart blood was collected and plasma TAC values were determined. In the therapeutic study, mice were ip injected with 10 mg FeDex together with DI, NAC (100 mg/kg) and PRBF (5 mg AE/kg) for 14 d. Mice were then sacrificed, their heart blood was collected and plasma TAC values were determined. Data are expressed as mean ± SD values (*n* = 6). ^d^
*p* < 0.05 when compared with DI alone in the therapeutic study. Abbreviations: AE = anthocyanin equivalent, PRBF = plasma-treated Riceberry™ rice flour, DI = deionized water, FeDex = iron dextran, NAC = N-acetylcystein, SD = standard deviation, TAC= total antioxidant capacity.

**Figure 5 molecules-26-04978-f005:**
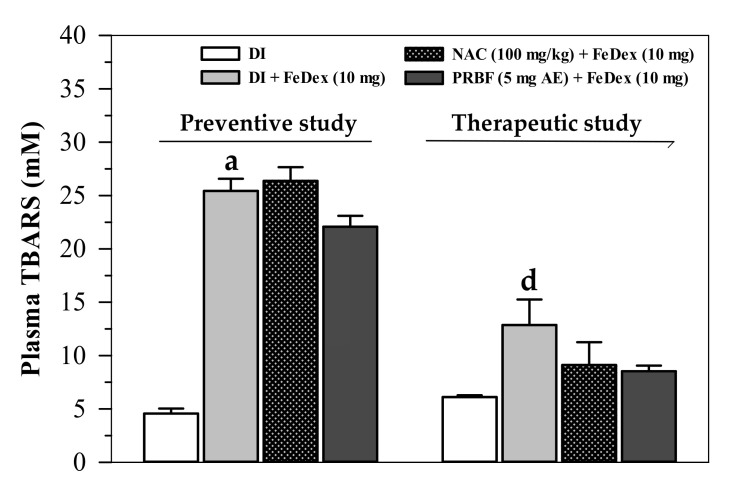
Levels of TBARS in the plasma collected from FeDex-loaded mice that had been treated with DI, NAC and PRBF. In the preventive study, mice were treated with DI, NAC (100 mg/kg) and PRBF (5 mg AE/kg) for 14 d. This was followed by a single ip injection with 10 mg FeDex after the last oral dose was administered on day 14. Mice were then sacrificed; their heart blood was collected and TBARS concentrations in the plasma were determined. In the therapeutic study, mice were ip injected with 10 mg FeDex together with DI, NAC (100 mg/kg) and PRBF (5 mg AE/kg) for 14 d. Mice were then sacrificed, their heart blood was collected and plasma TBARS concentrations were determined. Data are expressed as mean ± SD values (*n* = 6). ^a^
*p* < 0.05 when compared with DI alone in the preventive study; ^d^
*p* < 0.05 when compared with DI alone in the therapeutic study. Abbreviations: AE = anthocyanin equivalent, PRBF = plasma-Riceberry™ rice flour, DI = deionized water, FeDex = iron dextran, NAC = *N*-acetylcystein, SD = standard deviation, TBARS = thiobarbituric acid-reactive substances.

**Figure 6 molecules-26-04978-f006:**
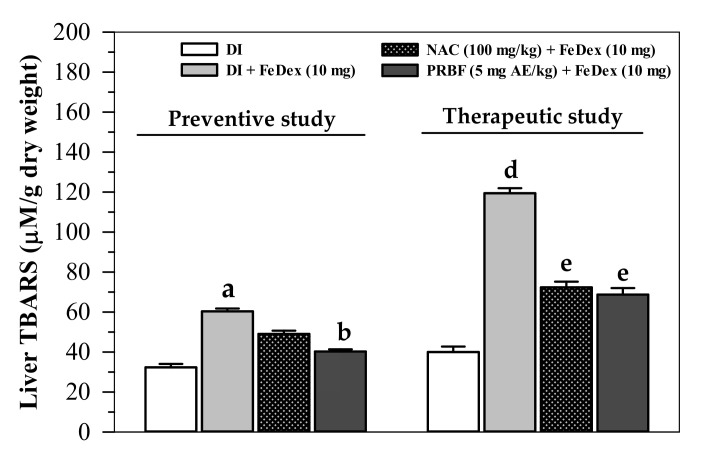
Levels of TBARS in the livers collected from FeDex-loaded mice that had been treated with DI, NAC and PRBF. In the preventive study, mice were treated with DI, NAC (100 mg/kg) and PRBF (5 mg AE/kg) for 14 d. This was followed by a single ip injection with 10 mg FeDex after the last oral dose was administered on day 14. Mice were then sacrificed; their internal organs were dissected and liver TBARS concentrations were determined. In the therapeutic study, mice were ip injected with 10 mg FeDex together with DI, NAC (100 mg/kg) and PRBF (5 mg AE/kg) for 14 d. Mice were then sacrificed, their internal organs were dissected and TBARS concentrations in the livers were determined. Data are expressed as mean ± SD values (*n* = 6). ^a^
*p* < 0.05 when compared with DI alone in the preventive study and ^b^
*p* < 0.05 when compared with DI in FeDex-loaded mice; ^d^
*p* < 0.05 when compared with DI alone and ^e^
*p* < 0.05 when compared with DI in FeDex-loaded mice in the therapeutic study. Abbreviations: AE = anthocyanin equivalent, PRBF = plasma-treated Riceberry™ rice flour, DI = deionized water, FeDex = iron dextran, NAC = *N*-acetylcystein, SD = standard deviation, TBARS = thiobarbituric acid-reactive substances.

## Data Availability

The authors confirm that the data supporting the findings of this study are available within the article.

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
