# Peer review of "Antioxidant Effects of Anthocyanin-Rich Riceberry™ Rice Flour Prepared Using Dielectric Barrier Discharge Plasma Technology on Iron-Induced Oxidative Stress in Mice"

_molecules, 2021, doi:10.3390/molecules26164978_

Round 1
Reviewer 1 Report
The manuscript entitled “Antioxidant effects of anthocyanin-rich Riceberry rice flour using dielectric barrier discharge plasma technology on iron-induced oxidative stress in mice”, authored by Natwalinkhol Settapramote and colleagues, deals with the investigation of the effects of dielectric barrier discharge (DBD) plasma technology in the preparation of Riceberry rice flour (PRBF) on iron-induced oxidative stress in mice.
The article contains really interesting data and I don’t have major concerns. However, some considerations must be made before considering it publishable on Molecules.
Keywords should be words not contained in the title, at most present in the abstract. Their usefulness is to make easier the searching of the article using the common scientific search engines. Since several keywords are already present in the title, and/or repeated several times in the abstract, I strongly advise the authors to add other keywords. As journal guidelines for authors clearly report, authors can provide up to 10 different keywords.
Authors should carefully check the style of the manuscript. In particular, the legend of the figures and tables should be reported according to the indications of the Journal.
The main focus of the authors in the revised manuscript are anthocyanin compounds. However, in the introduction section, too much general information is reported for these compounds. In particular, the authors should better explain that these compounds are widely but differently distributed in plants, and that they are important plant-specific markers (doi.org/10.3390/agriculture11030212).
Moreover, since the authors have performed quantitative analyses on anthocyanins, the results obtained can compared with those reported in this review (doi.org/10.3390/agriculture11030212) in which a cluster analysis of the diffusion and content of anthocyanins in the plant kingdom was carried out. In particular, the authors could affirm that the Riceberry extract investigated in their manuscript ranks among the 14 plant raw materials with the highest anthocyanin content.
Please consider to arrange the panels of the figures horizontally.
In the table, one-way ANOVA should be performed, and the letters of significance added as apex to the values.
Please refer to the journal guidelines to find a suitable Data Accessibility statement for the manuscript.
Reviewer 2 Report
The authors set out to investigate the effects of Riceberry rice flour produced using plasma technology on the oxidative balance of mice with iron overload (FeDex treatment). The authors compared the effects of PRBF with that of NAC. Two protocols were used, a preventive and a therapeutic. FeDex increased WBC count and caused a very discrete alteration in SOD activity in RBC, which was reversed or avoided in PRBF-treated animals. FeDex caused a very discrete alteration in plasmatic TAC in the therapeutic experiment. TBARS levels were clearly elevated in response to FeDex treatment; PRBF treatment mitigated this effect. Overall, the manuscript is well written, but English language revision is required; there are many grammar issues and odd terminology throughout the text. Figure legends and table titles are informative. However, at many instances it is not clear what is a statistically significant difference and what is an interpretations of the authors. Lastly, there are several statements not supported by data.
- The introduction must be reframed. In its current form it only highlights the outdated view of ROS (harmful) versus antioxidants (beneficial), which is currently known as a false dichotomy. ROS and other reactive species are fundamental for living systems, without them organisms would perish. On the other hand, too much antioxidant might be detrimental. There are numerous examples in the literature. Moreover, in the discussion, the authors should abandon such an outdated view and consider the role of vegetal extracts/compounds as cell signaling molecules and not simply radical scavengers (see: https://pubmed.ncbi.nlm.nih.gov/23747930/).
- The authors seem to neglect the chelating effect of many vegetal extracts. Would not PRBF contain molecules that could bind Fe? Would not that affect the redox status of animals?
- The authors seem to neglect the role of enzymes that act in the metabolization of lipid peroxidation products, such as aldehyde dehydrogenases and GSTs. The activity of these enzymes affect TBARS levels.
- The authors must explicitly state if they have compared NAC+Fe and PRBF+Fe groups against DI. In its current form, the reader is unsure if there is no difference or if the groups were simply not compared. Moreover, only by doing such comparison and finding significant difference the authors will support statements such as “increase in WBC was reversed by the NAC and PRBF treatments” and “only PRBF (5 mg AE/kg) treatment remarkably lowered the increase 203 in plasma TBARS levels”.
- To improve the readability and objectivity of the manuscript and considering that there were not statistically significant changes (as presented in figures 2 and 3), I suggest the authors to present data for BW and OWI in a supplementary material.
- Line 158: The statement that “PRBF (5 mg AE/g) treatments were more effective in restoring SOD activity” is not supported by data shown in figure 4. Please revise. The difference seems narrow. Regardless, the only way to support this statement would be a statistically significant difference between NAC and PRBF groups, which is not the case.
- Lines 102–106 and other several places: What do authors mean with “slightly lower”, “remarkably lowered” and “a bit less”? Were these alterations statistically significant? If not, please revise the text accordingly.
- Why the authors choose to measure SOD activity in erythrocytes? What is its biological meaning?
- Line 138: Was this change in PLT levels statistically significant? If not, please revise accordingly.
- Line 161: This kind of statement belongs to the discussion. Moreover, the authors should define what they mean with “antioxidant properties”. For example, hydrogen peroxide treatment can elicit the upregulation of many antioxidant enzymes. Is it an antioxidant?
- Line 304–305: Please provide additional primary evidence to support the idea that WBC number is an indicator of oxidative stress.
Minor issues:
- The authors should aim to reduce prolixity. For example, “were found to have increased” can be simply put as “increased” (without any loss of meaning), or “were not found to be significantly different” can be replaced with “did not differ”. Other expression, such as “According to our findings,” can be simply deleted.
- Some figure legends appear to have text in different fonts.
- Table 1 is a figure/image and not a properly formatted table.
- “Fantastically” and other adjectives/adverbs denoting subjective perceptions and opinions should be avoid in scientific texts.
- I would use only the term “preventive” consistently throughout the text instead of “protective”.
- Since methods are presented only at the end of the manuscript, please include the wavelength of the chromatogram in figures 1 in its legend.
- “loaded” is not an appropriate term to refer to animals treated with a drug.
Round 2
Reviewer 2 Report
The authors have addressed all issues raised by me in the previous round of review.